# Bioengineering of the Marine Diatom *Phaeodactylum tricornutum* with Cannabis Genes Enables the Production of the Cannabinoid Precursor, Olivetolic Acid

**DOI:** 10.3390/ijms242316624

**Published:** 2023-11-22

**Authors:** Fatima Awwad, Elisa Ines Fantino, Marianne Héneault, Aracely Maribel Diaz-Garza, Natacha Merindol, Alexandre Custeau, Sarah-Eve Gélinas, Fatma Meddeb-Mouelhi, Jessica Li, Jean-François Lemay, Bogumil J. Karas, Isabel Desgagne-Penix

**Affiliations:** 1Department of Chemistry, Biochemistry and Physics, Université du Québec à Trois-Rivières, 3351 Boulevard des Forges, Trois-Riviere, QC G9A 5H7, Canada; 2Groupe de Recherche en Biologie Végétale, Université du Québec à Trois-Rivières, Trois-Riviere, QC G9A 5H7, Canada; 3Department of Biochemistry, Schulich School of Medicine and Dentistry, Western University, London, ON N6A 5C1, Canada; 4Centre National en Électrochimie et en Technologies Environnementales Inc., 2263 Avenue du Collège, Shawinigan, QC G9N 6V8, Canada

**Keywords:** diatom, metabolic engineering, olivetolic acid cyclase, tetraketide synthase, synthetic biology, cannabinoids, microalgae

## Abstract

The increasing demand for novel natural compounds has prompted the exploration of innovative approaches in bioengineering. This study investigates the bioengineering potential of the marine diatom *Phaeodactylum tricornutum* through the introduction of cannabis genes, specifically, tetraketide synthase (TKS), and olivetolic acid cyclase (OAC), for the production of the cannabinoid precursor, olivetolic acid (OA). *P. tricornutum* is a promising biotechnological platform due to its fast growth rate, amenability to genetic manipulation, and ability to produce valuable compounds. Through genetic engineering techniques, we successfully integrated the cannabis genes *TKS* and *OAC* into the diatom. *P. tricornutum* transconjugants expressing these genes showed the production of the recombinant TKS and OAC enzymes, detected via Western blot analysis, and the production of cannabinoids precursor (OA) detected using the HPLC/UV spectrum when compared to the wild-type strain. Quantitative analysis revealed significant olivetolic acid accumulation (0.6–2.6 mg/L), demonstrating the successful integration and functionality of the heterologous genes. Furthermore, the introduction of TKS and OAC genes led to the synthesis of novel molecules, potentially expanding the repertoire of bioactive compounds accessible through diatom-based biotechnology. This study demonstrates the successful bioengineering of *P. tricornutum* with cannabis genes, enabling the production of OA as a precursor for cannabinoid production and the synthesis of novel molecules with potential pharmaceutical applications.

## 1. Introduction

Over a thousand of the molecules produced in *Cannabis sativa* plants have been reported in the literature, including more than a hundred cannabinoids (CBs) [1]. The antinociceptive and appetite-stimulating properties of CBs have been studied thoroughly, as well as their association with relieving the symptoms associated with different diseases such as multiple sclerosis and the relief of chemotherapy side effects [2,3,4,5]. The most studied CBs are ∆9-tetrahydrocannabinol (THC), which has psychoactive potential, and cannabidiol (CBD), which acts on human endocannabinoid receptors to modulate pain and is already contained in many pharmaceutical products available on the market, such as Sativex and Bediol [6,7,8]. Other therapeutic properties of specific CBs that could be administrated in a purified form are still under investigation and offer a promising alternative for different human pathology applications [5]. In planta, phytocannabinoids are produced in glandular trichomes via fatty acid and terpenoid precursors [9]. The genes and enzymes implicated in the cannabinoid pathway were characterized by several plant biologists shortly after the sequencing of the *C. sativa* genome [9,10,11,12]. The first step in the CB biosynthetic pathway is the formation of olivetolic acid (OA) from malonyl-CoA and hexanoyl-CoA, through a two-step reaction performed by a type-III polyketide synthase named tetraketide synthase (*Cs*TKS) followed by a cyclase named olivetolic acid cyclase (*Cs*OAC), both of which are localized in the cytosol or cells from the glandular trichomes (Figure 1). The enzymes belonging to the type-III polyketide synthase (PKS) superfamily are responsible for producing a wide array of central frameworks that are found in specialized plant metabolites of significant medicinal value. These metabolites encompass various categories, such as flavonoids, stilbenes, chromones, pyrones, phloroglucinols, resorcinols, xanthones, acridones, and quinolones [13,14]. Examples of PKS include chalcone synthase, which accepts *p*-coumaroyl-CoA as the starter substrate to catalyze three successive condensations with malonyl-CoA to generate naringenin chalcone [15,16] and stilbene synthase, which generates resveratrol [17]. The polyketide synthase from *C. sativa* (also named tetraketide synthase or olivetol synthase) was characterized and has been shown to be dependent on OAC for the formation of OA (Figure 1) [9]. The 3D structure of *Cs*OAC is most accurately described as a dimeric α + β barrel (DABB) protein, with similarities to other plant DABBs, especially those implicated in stress responses and to the DABB proteins found in *Streptomyces* bacteria [9]. The aromatic precursor OA is a resorcinol carrying an alkyl chain (named alkylresorcinolic acid), which is condensed with a monoterpenoid moiety (geranyl diphosphate, GPP) to yield the general structure of the most well-known CBs, as reported in feeding studies of 13C-labeled glucose in the early 1960s [18,19]. Although not all CBs are derived from the OA precursor, it remains the most widely characterized pathway in *C. sativa* [20]. The prenyl moiety of GPP is transferred to OA by an aromatic prenyltransferase (*Cs*APT) to form cannabigerolic acid (CBGA) (Figure 1). This core intermediate is the backbone of the remaining reaction, which diverges to form cannabidiolic acid (CBDA) and tetrahydrocannabinolic acid (THCA) through CBDA-synthase (CBDAS) and THCA-synthase (THCAS), respectively (Figure 1), as well as other CBs such as cannabichromenic acid (CBCA) via cannabichromenic acid cyclase (CBCAS) [9,11,21,22]. The neutral phytocannabinoids THC and CBD are generated from cannabinolic acids by non-enzymatic decarboxylation and are often produced upon exposure to light or heat (Figure 1) [21]. Aside from CBs, the cannabis plants also produce other specialized metabolites from shared precursors with cannabinoids, such as fatty acids, competing with olivetolic acid synthesis, and terpenoids such as limonene [23], competing for the GPP pool [24,25].

The yield of CB in plants is limited, both in quality and in quantity; the maximum production is due to a mixture of these metabolites accumulating in the flowers (20 mg/g fresh weight) [6,26]. In addition, the lack of scientific literature on cannabis physiology leads to the absence of standardized culturing protocols, resulting in a variation in CB profiles, quality, and quantity. In addition, cannabis pathogens such as the gray mold causal agent (*Botrytis cinerea*) affect the overall plant yield and metabolite composition [27]. As well as multiple unknown facets of CB production in planta, this suggests that the implementation of a heterologous expression system with fewer components and fewer competing enzymes might be a solution for achieving a higher production yield of single CB components, such as CBD or THC. Therefore, synthetic biology offers an important alternative method to produce CBs for the pharmaceutical field with less risk and fewer crop management challenges.

Several research groups aimed to design de novo or improve pre-existing cannabinoid production in plant tissue cultures, but apart from the transformation rates, in vitro culture viability and yield are still to be optimized [6]. For instance, few groups succeeded in transforming yeast (*Saccharomyces cerevisiae*) and bacteria (*Escherichia coli*) to produce the different metabolites of the CB pathway, such as OA or CBGA [28,29]. The production of these metabolites in yeast and bacteria often involves the addition of costly precursors such as acetyl-CoA, hexanoyl-CoA, or olivetolic acid [28,30], or involves supplementing the media with sugars [31], such as galactose, to provide a more affordable alternative with a long series of gene stacking. For instance, the production of OA in *Dyctiostelium discoideum* on a 300-L scale led to 4.8 µg/L [32]. OA production in *E. coli* reached 80 mg/L after introducing a series of modifications to increase the CoAs pool [33]. However, the bacterial and yeast translation and post-translational mechanisms, as well as their metabolic chassis, are distinct from the mechanisms found in higher plants. This resulted in low yields, such as picomoles per unit of optical density, from these microorganisms when compared to the bioengineering and supplementation costs [31,34]. By contrast, diatoms are photosynthetic organisms that share the main metabolic paths with plants, making them promising candidates for the production of heterologous compounds [9]. All the precursor pathways, such as acetyl-CoA, hexanoyl-CoA, GPP, etc., are present in diatom cells at different times of the cell cycle, which reduces the need to supplement the culture media with costly precursors or sugars [35]. Also, the extrachromosomal transformation of diatoms via bacterial conjugation allows the fairly stable expression of heterologous genes [36,37]. Similar advances in the bioengineering of diatoms have been shown recently, allowing the production of monoterpenoids such as geraniol, an insect repellant, in the model diatom *Phaeodactylum tricornutum* [36].

Thus, the aim of this study is to produce the CB precursor OA in the marine microalga, *P. tricornutum*. Here, we inserted cannabis *Cs*TKS and *Cs*OAC genes into a stable optimized episome that was assembled in *S. cerevisiae* [38,39]. The recombinant episome was then transformed via a trans-kingdom conjugation between *E. coli* and *P. tricornutum* [38]. The transconjugant strains were screened for their ability to produce molecules of interest, using a high-performance liquid chromatograph coupled to an ultra-violet detector (HPLC-UV). We detected OA (0.6–2.6 mg/L) in *P. tricornutum* transconjugants that issued from two different expression cassettes. Although the production of the OA compounds of interest was temporary, this work sheds light on a powerful and suitable system for plant metabolite heterologous production in a cost-effective phototropic system employing *P. tricornutum*.

## 2. Results

### 2.1. Heterologous Protein Expression and Localization in the Diatom P. tricornutum

#### 2.1.1. Transformation Validation and Transconjugant Characterization

To inform and design the metabolic engineering strategy, we first performed computational analyses to assess the availability of the lipid-derived precursors required to produce OA and cannabinoids in *P. tricornutum* with in silico tools and showed that upstream pathways existed in the host organism. According to the Biocyc database [40], hexanoyl-CoA can be formed in *P. tricornutum* in the peroxisome via the fatty acid β-oxidation pathway, mainly by long-chain acyl-CoA synthase (*Pt*ACS3 and *Pt*ACS4), while malonyl-CoA is formed via the activity of an ACYL-CoA carboxylase (ACCase). Moreover, the geranyl pyrophosphate (GPP) pool is present at detectable levels in *P. tricornutum* [36].

To produce OA in *P. tricornutum*, three different expression cassettes were designed using p*Pt*GE30 as a backbone plasmid DNA (Figure 2A), namely, *Pt*OA1, *Pt*OA2, and *Pt*OA3, with various combinations of *C. sativa* genes encoding for *Cs*TKS and *Cs*OAC (Figure 2B). Each expression cassette was codon-optimized for *P. tricornutum* and assembled (Appendix A, Figure A1, Table A1, Table A2 and Table A3) according to the HIVE lab codon usage table (CUT) [41].

The transconjugant strains *Pt*OA1, *Pt*OA2, and *Pt*OA3 did not present significant differences in growth and shape compared to *P. tricornutum* when transformed with the p*Pt*GE30 empty vector or wild-type strains (Figure 3A–C). In addition, morphological properties such as morphotype, size, granulosity, and chlorophyll levels did not significantly differ between the studied strains, as evaluated by microscopy (Figure 3A) and flow cytometry (Figure 4).

#### 2.1.2. Heterologous Protein Detection and Localization

The successful expression of cannabis genes and the accumulation of the corresponding enzymes was observed in *P. tricornutum* transconjugants. Indeed, the accumulation of the *Cs*TKS-T2A (45 kDa band) protein in *Pt*OA1C1 (Figure 5A), a 68 kDa uncleaved protein corresponding to CsTKS-His-T2A-CsOAC-cMyc in *Pt*OA2C2 (Figure 5B), and an 85 kDa band corresponding to YFP-CsTKS-3(GGGGS)-CsOAC-cMyc in *Pt*OA3C1 (Figure 5C) was detected via Western blot analysis in PCR-positive *P. tricornutum* transconjugants (Appendix A, Figure A1B). However, the construction with the self-cleavable sequence did not yield detectable successful single proteins in *Pt*OA2. This could be caused by the tag (6xHis) used in the OA2 construct but not in the OA1 construct, changing the efficiency of cleavage (Figure 5B). Then, we studied the localization of *Cs*TKS in *PtOA*3, taking advantage of the presence of the fluorescent YFP fusion protein. We observed under confocal microscopy that the heterologous recombinant protein (YFP-*Cs*TKS-3(GGGGS)-CsOAC-cMyc) produced in *PtOA*3 accumulated and was localized to the cytosol (Figure 5D).

### 2.2. Temporal Metabolite Production

#### 2.2.1. Metabolite Detection and Identification

Selected positive transconjugant strains, displaying protein accumulation via Western blot analysis, were cultivated, extracted, and used for targeted metabolite analysis. OA was detected at 2.6 mg/kg in *Pt*OA1 and 0.54 mg/kg in *Pt*OA2, using HPLC-UV, whereas no peaks at the OA retention time were detected in the *Pt*OA3 extracts (Figure 6 and Appendix A, Figure A3).

Interestingly, additional peaks with similar retention times to the cannabinoid standards of CBN, CBD, and THC, and their corresponding acid forms were detected in the extracts of OA-producing transconjugants (*Pt*OA1C1 and *Pt*OA2C2) (Appendix A, Figure A4). Liquid chromatography–mass spectrometry (LC-MS) analyses using electrospray ionization in the positive mode (ESI+) were performed to validate if the mass-to-charge ratio (*m*/*z*) of these peaks corresponded to protonated CBs (M + H)^+^. The compounds were first scanned for specific masses corresponding to the metabolites of interest, then, in the case of a positive scan, another LC-MS analysis was performed to make sure that the fragments corresponded to the target compound. Unexpectedly, the daughter *m*/*z* obtained after employing collision-induced energy (CID) at 40 V of the selected RT and *m*/*z* corresponded to the cannabinoid standards (Appendix A, Figure A4). This suggests that *P. tricornutum* may contain endogenous enzymes that are capable of converting OA into CB-like metabolites such as CBGA, CBDA, and THCA. Such enzymes could be mimicking the activity of cannabis APT (using the endogenous pool of GPP) and CBDAS/THCAS (Figure 1). Candidate enzymes from the diatom database were searched and low-similarity sequences for APT and THCAS were identified (Appendix B, Figure A6). However, further characterization is needed and might help find closer candidates among microalgae. Alternatively, candidates from a completely different type of enzyme could be involved.

To test for the presence of such endogenous enzymes, we performed a supplementation assay with *P. tricornutum* transconjugants harboring the p*Pt*GE30, which were devoid of cannabis genes (Table 1). We first added OA and GPP for 16 h and could subsequently detect small amounts (circa 3–5 mg/kg) of CBGA and other CB-like metabolites using HPLC analysis. This suggests the presence of an APT-like enzyme in *P. tricornutum* (Table 1). Next, we supplemented the sample with CBGA and detected metabolites that behaved in a similar way to CB on an HPLC chromatogram, but we were unable to confirm their identity (not all were appropriate *m*/*z* fragments) via LC-MS.

#### 2.2.2. Metabolite Production Timeline

Olivetolic acid and cannabinoid-like metabolites were detected up to three months after obtaining and subculturing each transconjugant in optimal conditions with no supplementation of any kind. However, a gradual decrease in the concentration of the OA and CB-like metabolites was observed. After six months, the transconjugants completely lost the ability to produce these desired metabolites. Along with production loss, we noticed that the *P. tricornutum* transconjugant morphotype also changed from triradiate (Figure 3A) to fusiform (Figure 5D), and that the initial triradiate morphotype of the transconjugants was never restored.

## 3. Discussion

The bioengineering of *P. tricornutum* with cannabis genes represents a promising approach to addressing the challenges associated with traditional cannabinoid production. By harnessing the unique capabilities of marine diatoms, this study paves the way for sustainable and controlled cannabinoid biosynthesis. Furthermore, the successful expression of cannabis genes in a non-native host highlights the versatility of synthetic biology in the context of metabolic engineering. This work represents the first report of cannabinoid engineering in brown microalgae and provides a proof-of-concept evaluation of *P. tricornutum* for the heterologous production of OA and cannabinoids. We provide evidence that the natural metabolism of the marine diatom *P. tricornutum* is well-suited for use in metabolic engineering to produce the cannabinoid precursor, OA, and CB-like metabolites.

The episomal vectors, *Pt*OA1 and *Pt*OA2, contained both *CsTKS* and *CsOAC* genes, separated by a self-cleaving peptide sequence and under the control of the same promoter, to ensure similar levels of expression of both genes and, subsequently, comparable levels of accumulated enzymes for performing the two-step reactions. The transconjugants *Pt*OA1 and *Pt*OA2 only differed in terms of the addition of a 6xHis tag on *Cs*TKS for easier detection of the protein. In *Pt*OA3, both genes were linked together with a linker and tagged with YFP for visualization and localization studies.

Although a previous study demonstrated the occurrence of episomal rearrangements [42], in this study, the *P. tricornutum* transconjugant strains obtained for each episomal vector did not show significant DNA sequence rearrangement after 1 year (Appendix A, Figure A1 and Table A4). This suggests that episomal rearrangements could be sequence- or DNA-motif-specific. Also, the *Pt*OA1-3 transconjugant strains grew similarly to the WT and EV strains (Figure 3 and Figure 4). This suggests that the presence and expression of the transgenes and the produced metabolites did not significantly affect the growth and division of *P. tricornutum*. Genome analysis of green algae, diatoms, and higher plants revealed similarities in lipid metabolism [43,44]. Therefore, the fact that there is little to no effect of *Cs*TKS and *Cs*OAC expression on *P. tricornutum* growth and fitness could be a result of the nature of the pathway, as an extension of fatty acid metabolism that is fairly close to the diatom chassis, and a preference for energy conversion into lipids.

Heterologous protein detection was not consistent among the different transconjugant strains (Figure 5 and Appendix A, Figure A2). For example, the four analyzed transconjugant strains of *Pt*OA1 show a similar pattern of accumulation, whereas, in the *Pt*OA2 strains, the T2A sequence did not appear to be cleaved. This could be explained either by the possible presence of negatively acting *cis* elements in the sequence of *Pt*OA2 transgenes [43] or by the nature of the amino acid environment that surrounds the T2A sequence cleavage site [45]. It was also reported that the gene following the T2A sequence would be affected on the expression level [46] and, thus, would yield fewer *Cs*OAC transcripts, but the effect of this phenomenon would be validated if the C10–C12 polyketide product was available as a standard for HPLC detection, which is not the case.

When compared to previous studies, the production of OA in *Yarrowia lypolytica* slightly changed when *Cs*TKS and *Cs*OAC were expressed under different promoters, compared to when the two genes were fused [42]; in *P. tricornutum*, the *Pt*OA1 showed a higher quantity of OA (Figure A3) compared to there being none in *Pt*OA3 when the two genes were fused by a glycine-serine linker. This is not applicable in *Pt*OA2 since the Western blot did not show cleaved proteins, which could indicate that the cleaved proteins were below detection levels.

To our knowledge, this is the first time that a brown microalgae host has demonstrated the capacity to accumulate OA and use it for the synthesis of CB-like metabolites. The heterologous production of OA was observed in the different transconjugants of *P. tricornutum*, transformed with episomal constructions and harboring various combinations of *C. sativa TKS* and *OAC* genes (Figure 2; Appendix A, Figure A1 and Table A2). The cassette designs in this study were fairly simple and direct, with only two main cannabis genes being introduced. When compared to the literature, the first approaches regarding yeast achieved the production of only 0.48 mg/L of OA by expressing *Cs*TKS and *Cs*OAC, and by feeding the culture with sodium hexanoate [9]. Here, we obtained higher yields (0.56–2.6 mg/kg) compared to yeast with and without hexanoate supplementation [9]. In *Y. lipolytica*, it was bioengineered with *Pseudomonas* sp. *LvaE*, encoding a short-chain acyl-CoA synthetase, acetyl-CoA carboxylase, pyruvate dehydrogenase bypass, NADPH-generating malic enzyme, as well as the activation of the peroxisomal β-oxidation pathway and the ATP export pathway to redirect the carbon flux toward OA synthesis [47]; the yield was also less than 0.5 mg/L until batch culture optimization.

Further optimization of the *P. tricornutum* transconjugant strains by boosting the pool of precursors could increase the production of OA. In *E. coli*, besides introducing *Cs*TKS and *Cs*OAC, metabolic engineering consisted of the overexpression of the acetyl-CoA carboxylase subunits A, B, C, and D from the same host to ensure the high production of malonyl-CoA, and a reversal of beta-oxidation that allowed the accumulation of hexanoyl-CoA [33]. This led to a higher quantity of OA (46.3 mg/L) after the optimization of culture conditions and induction parameters [33]. In *P. tricornutum*, hexanoyl-CoA and malonyl-CoA occur naturally, due to the long-chain acyl-CoA synthase (*Pt*ACS3 and *Pt*ACS4) for hexanoyl-CoA and to the acetyl-CoA ATP-dependent carboxylase (*Pt*ACCase) for malonyl-CoA [43]. *P. tricornutum* was also shown to accumulate intracellular amounts of GPP, along with neryl diphosphate (NPP) [36,48]; both precursors are donors of the prenyl moiety of several cannabinoids, such as CBGA and cannabinerolic acid (CBNA) in cannabis [46]. However, overexpression of these upstream genes could eliminate the need for supplementation while maintaining the necessary pools of precursors for more stable production of cannabinoids, which is similar to the process used with yeast [31,34].

Along with precursor bioavailability, endogenous enzymes are required to enable the conversion of OA into CB or CB-like compounds, as observed in this study, either by direct production (Figure A4) or by supplementation assays (Table 1). Bioinformatic analyses led to the identification of two putative enzymes (Appendix B, Figure A6) that are encoded by Phatr3_J37858 and Phatr3_J2738.t1 and are capable of prenyltransferase activity, thereby allowing the production of CBGA. Both enzymes are UbiA prenyltransferase family members that could be used originally by the diatom to produce terpenes or flavonoids [49,50]. The in silico identification of THCAS and CBDAS-like endogenous candidate genes is more intricate, given the specificity of these enzymes. Our approach can also be used to further characterize endogenous diatom enzymes with multiple functions by testing the reactivity of diatoms to drug precursors in higher-level plants.

The episomal sequences were well-conserved in the timeframe of this study in all three constructions of interest (in DNA and proteins). However, after such a timeframe, the production of OA was not detectable, perhaps due to silencing via an epigenetic mechanism [42], mRNA or protein stability [37], and, less probably, episomal rearrangements, as seen in other cases [42]. It would be useful to characterize a larger number of transformed *P. tricornutum* using a method similar to the process used with yeast [31,34], or to try randomly integrated chromosomal expression (RICE). Interestingly, a loss of production was observed in parallel with a change in cell morphotype from a triradiate to a fusiform population, which could also be associated with changes in the intracellular levels of certain precursors related to general metabolism [51]. A loss of heterologous production of specialized metabolites was previously observed for *E. coli*, where, after the 31st generation, lycopene production decreased from 3.3% of its initial yield [52] due to segregational plasmid instability. Similarly, the use of cyanobacteria as a heterologous production platform faces many challenges, such as a loss of productivity due to the burden or loss of the metabolic pathway and genetic instability [52,53]. In yeast, low expression-cassette instability in large fermentation batches led to a decrease in recombinant protein production [54]. Taken together, these studies show that a loss of production seems to occur elsewhere in other systems, as a result of different underlying problems. In addition, these further demonstrate that like other systems, diatoms need further optimization of the metabolic pathway-encoding constructs discussed below to ensure a suitable platform for the heterologous production of specialized metabolites. Recently, cloning the *Metarhizium anisopliae ARSEF23* highly reducing PKS enzyme (*MaOvaA)*, in addition to a non-reducing PKS (*MaOvaB*) and a domain PKS (*MaOvaC)* consisting of an acyl carrier protein and a thioesterase, in the fungus *Aspergillus nidulans* yielded high titers of OA (80 mg/L) [55]. Thus, it would be interesting to compare the expression and behavior of cannabis enzymes regarding fungal enzymes such as highly reducing and non-reducing PKS, along with thioesterase and acyl carrier enzymes [50]. Another approach to increase OA titers could be the construction of alternative pathways using enzymes found in other OA-producing plants than cannabis, in order to achieve OA and other CB precursors while using fungal enzymes or mutation-enhanced enzymes. It is possible, from an evolutionary point of view, that other sources of transgenes are more suitable for heterologous expression in diatoms.

Moreover, in this study, low OA yield could be explained by the detection of cannabinoid-like compounds in higher titers (Figure A4) similar to those observed in other studies, where flavonoid precursors were consumed to form flavonoids in bioengineered yeast [56]. Further steps of compartmentalization of the products would be an interesting strategy, given the possible cytotoxicity of the cannabinoids in many systems [12,57].

The application of *P. tricornutum* as a production platform for OA raises intriguing questions about its metabolic compatibility with the broader CB biosynthesis pathway. Further research is warranted to understand how the engineered diatom handles the downstream conversion of OA into various cannabinoids. Additionally, optimizing the growth conditions, such as light intensity, nutrient availability, and cultivation scale, will be crucial for maximizing production yields [51,58,59]. The successful synthesis of olivetolic acid in *P. tricornutum* opens the door to a range of future possibilities. Further pathway engineering could enable the production of specific compounds, offering a renewable and controllable source of cannabinoids for medical and industrial applications. The engineered diatom strains could be integrated into larger bioprocessing systems, harnessing their natural growth advantages for cost-effective production.

## 4. Materials and Methods

### 4.1. Microbial Strains and Growth Conditions

*Saccharomyces cerevisiae* VL6-48 (ATCC MYA-3666: MATα his3-Δ200 trp1-Δ1 ura3-52 lys2 ade2-1 met14 cir0) was used for the yeast assembly, as described previously [38]. Positive yeast strains containing the His selection were grown on minimal yeast media without histidine. A pool of the grown yeasts was harvested 5 days after assembly and the total DNA was extracted, as described previously [60]. The assembled plasmids were then transformed via electroporation and amplified in *Escherichia coli* (Epi300, Epicenter, Biosearch Technologies, Guelph, ON, Canada), which was grown on Luria Broth (LB) media supplemented with the appropriate antibiotic chloramphenicol (25 mg·L^−1^) overnight at 37 °C. The plasmids were then extracted from the chloramphenicol-*E. coli* colonies, which were tested by cPCR using a miniprep kit, allowing the extraction of large vectors (EZ10 DNA miniprep kit, Bio Basic Inc., Markham, ON, Canada).

### 4.2. Diatom Transformation, Selection, and Subculturing

The plasmids were verified by next-generation sequencing and were then amplified in an *E. coli* Epi 300 strain containing the pTA-MOB plasmid to enable conjugation with wild-type diatoms, as described in the literature [38]. *P. tricornutum* (Culture Collection of Algae and Protozoa, CCAP 1055/1), was kindly provided by Prof. Bogumil Karas. The transfer of plasmid DNA to *P. tricornutum* via conjugation from *E. coli* was performed as described by Karas et al. [38]. For this process, 250 μL of wild-type *P. tricornutum* culture was adjusted to a density of 10^8^ cells/mL, while the *P. tricornutum* cell density was obtained by plating 1 mL of wild-type *P. tricornutum* on 1/2 × L1 1% agar plates and growing them at 18 °C under cool fluorescent lights (75 μE m^−2^s^−1^) on a light/dark cycle of 16/8 h for 4 days. Prior to transformation, 1 mL of L1 media was added to each agar plate, and the cells were first scraped and then harvested via pipetting in a sterile tube. The cells were then diluted and mounted in an improved Neubauer hemacytometer (BLAUBRAND^®^ counting chamber, MilliporeSigma Canada Ltd., Oakville, ON, Canada) to be counted, then the cell concentration was adjusted to 5.0 × 10^8^ cells mL^−1^. A volume of 50 mL of *E. coli* culture containing the assembled and pTA-MOB plasmids was grown at 37 °C under agitation to reach an OD_600_ of 0.9, then the volume was centrifuged at 3000× *g* for 10 min and resuspended in 500 μL of SOC media. Conjugation was initiated by adding 200 μL of *P. tricornutum* to 200 μL of *E. coli* cells. The cell mixture was then plated on 1/2 L1, 5% LB, 1% agar plates, incubated at 30 °C for 90 min in the dark, and then transferred to 18 °C in the light and grown for 48 h. Two days later, 1 mL of L1 media was added to the plates and the cells were collected by scraping, then a volume of 200 μL of cells was plated on 1/2 L1, 1% agar plates supplemented with zeocin at 50 μg mL^−1^ for selection and were incubated at 18 °C under light (75 μE m^−2^s^−1^). Two weeks later, positive colonies appeared and were streaked again on 1/2 L1, 1% agar plates supplemented with zeocin at 50 μg mL^−1^ for the verification of plasmid stability. The recombinant colonies of *P. tricornutum* were checked via cPCR. Four positive colonies were retained from each construct, from which the plasmids were extracted for further confirmation. The extracted vectors were then amplified in *E. coli* Epi300; these were extracted using the EZ10 kit and sequenced using next-generation sequencing via Illumina MiSeq technology at the Massachusetts General Hospital’s Center of Computational and Integrative Biology (MGH CCIB DNA Core, Boston, MA, USA). In parallel, transformed *P. tricornutum* culture was launched from a single colony until obtaining an OD_680_ nm of 0.06–0.1. The bioengineered *P. tricornutum* strains were then sub-cultured every 10 days in a 1/3 volume ratio of fresh L1 Si^−^ medium containing zeocin (50 mg·L^−1^).

### 4.3. Transgene Sequences Validation

*C. sativa* tetraketide synthase (*Cs*TKS) and olivetolic acid cyclase (*Cs*OAC) amino acid sequences were obtained from *C. sativa* through Genbank, under the respective accession numbers of ACD76855.1 and JN679224.1 (Appendix A, Table A1). The amino acid sequence was then converted into a DNA sequence with respect to *P. tricornutum*’s codon usage table, which is accessible through NCBI Genbank, as well as the HIVE laboratory’s codon usage table (CUT) [41]. Several cassettes containing the genes of interest were designed; in this article, we describe three of them. (1) *Pt*OA1 was a construction where TKS and OAC were linked by a self-cleavable sequence from the *Thosea asigna* virus (T2A) [45] without tags, to ensure a similar enzymatic site exposition as that seen in *C. sativa*. (2) *Pt*OA2 is a construction where TKS and OAC were tagged, respectively, in the C-terminal with 6xHis and c-Myc tags for possible protein detection or purification and were linked by a T2A sequence. (3) *Pt*OA3 is a construction designed to present TKS coupled with a reporter gene fluorescent protein eYFP on the N-terminal fused by (GlyGlyGlyGlySer)_3_ to OAC, which is tagged with c-Myc on the C-terminal. Each inserted construction was designed to be expressed under the strong constitutive promoter 40SRPS8 and the FcpA terminator in a p*Pt*GE30 backbone vector [61] that contained the zeocin resistance gene *ShBle* as a selection marker. The plasmid p*Pt*GE30 contains a centromeric yeast fragment, allowing it to remain extrachromosomal [38]. All gene sequences were codon-optimized for *P. tricornutum*’s optimal codon usage, with a GC content ranging from 48 to 55%. A representation of each plasmid design is detailed in Figure 1. The primers used for the gene amplification and DNA fragment assembly are listed in Appendix A, Table A1. The fragments and transgene sequences that have been optimized for higher expression in *P. tricornutum* are detailed in Appendix A, Table A2. The sequencing results of the plasmid DNA obtained from *P. tricornutum* clones described in this study are detailed in Appendix A, Table A3.

### 4.4. Heterologous Protein Detection

Protein detection in the positively transformed strains of *P. tricornutum* and the negative control (wild type, transformed with p*P*tGE30) was performed on 50 mL (OD_680_ nm 0.1 a.u.) cultures that were 6 days old with OD_680_ nm values between 0.9 and 1.7 a.u. The cells were centrifuged at 4000 rpm for 20 min at 4 °C. The resulting pellets were weighed and resuspended in loading buffer 1× (0.8% sodium dodecyl sulfate, 0.05 M Tris-HCl pH 6.8, 6% glycerol, bromophenol blue, and 3.2% β-mercaptoethanol) to a final concentration of 500 mg FW·mL^−1^. The cell lysates were boiled at 95 °C for 5 min and were then centrifuged at maximum speed for 5 min at room temperature. A volume of 35 µL of the supernatant was loaded into 12% SDS-PAGE gel and migrated at 80 volts until the proteins passed through to the stacking gel, then the voltage was raised to 120 volts. The gel was then transferred using the BioRad Trans-Blot Turbo Transfer system at 2.5 A for 15 min. The blot was equilibrated in 1× TBS solution and blocked with 5% milk for 1 h at room temperature, before washing it three times with TBST solution and adding the primary antibody overnight at 4 °C. In this study, the various primary antibodies used for the T2A sequence were purchased from Sigma Aldrich (cat. #ABS31, Boston, MA, USA), the 6X-His Tag Monoclonal antibody from Thermofisher (cat. #MA1-21315, Waltham, MA, USA), and anti-GFP/CFP/YFP from Cedarlane (cat. #CLH106AP, Burlington, ON, Canada), all at a 1:1000 dilution in 3% BSA. After three washes with TBST solution, the *Pt*OA1 sample blot was incubated for 1 h in a 1:20,000 dilution in 5% milk of the Immun-Star Goat Anti-Rabbit (GAR)-HRP conjugate from Bio-Rad (Mississauga, ON, Canada, cat. #1705046). Meanwhile, the *Pt*OA2 and *Pt*OA3 blots were incubated under the same conditions with Immun-Star Goat Anti-Mouse (GAM)-HRP conjugate from BioRad (cat. #1705047, Hercules, CA, USA). The multiple Tag protein (GenScript, cat. # M0101, Piscataway, NJ, USA) and the recombinant YFP purified protein (10 ng) from *E. coli* were used as the positive controls. After three washes with TBST solution, protein detection was performed using the Clarity Max Western ECL Substrate-Luminol solution from Bio-Rad (cat #1705062S, CA, USA). The blots and Coomassie-stained gels were visualized using a ChemiDoc Imaging System with Image Lab Touch software 2.4 (Bio-Rad, cat #12003153, Hercules, CA, USA) and Image Lab™ 5.2 software (Bio-Rad, cat # 1709690, Hercules, CA, USA).

### 4.5. Subcellular Localization of YFP

Live cell images of the four-day-old culture were captured using a Leica SP8 confocal laser microscope (Leica, Wetzlar, Germany) with an HCX PL APO 60×/1.25–0.75 Oil CS objective. The excitation of YFP and chlorophyll fluorescence occurred at 488 nm with a 65 mW argon laser. YFP fluorescence emission was detected between 552 and 560 nm, whereas chlorophyll fluorescence was detected at a bandwidth of 625–720 nm. Bright-field light microscopy images were also taken. The images were analyzed using the Fiji software (https://imagej.net/software/fiji/) for Windows (64-bit) [62].

### 4.6. Flow Cytometry Analysis

The BD FACS Melody device (BD Biosciences, La Jolla, CA, USA), equipped with blue (488 nm), red (640 nm), and violet (405 nm) lasers, and a Beckn Cytoflex FC500 equipped with Argon (488 nm) and HeNe (633 nm) lasers were used to measure YFP fluorescence on the FITC channel (527/32 and 525/15 nm) and chloroplast autofluorescence on the APC channel (660/10 nm and 675/15 nm, respectively). The *P. tricornutum* samples were diluted to an OD_680_ of 0.1 and were then analyzed at a fixed flow rate of 1 for at least 10,000 events per sample, conducted with 3 replicates. The diatoms were first gated using side-scattered light (SSC) versus forward scatter plots to determine the targeted population and were then gated using the chlorophyll levels (650 nm) (Figure 4). All data acquisition and analysis were carried out with the BD FlowJo version 10 software (BD Biosciences, La Jolla, CA, USA, 2020). All FACS experiments were conducted in triplicate.

### 4.7. Metabolite Extraction Method

The wild-type and positive transconjugant strains were pre-cultured in L1 medium, as mentioned previously. Approximately 100 mg of wet biomass from each culture was harvested by centrifuging 50 mL of culture at 1500× *g* and at 4 °C for 10 min. The pellets were resuspended in 5 mL of ethanol 95% (cat.# P016EA95), vortexed 5 times for 1 min each time, and stored at −20 °C overnight. The metabolite extracts were separated from the cells by centrifuging at 1500× g and at 4 °C for 10 min. The supernatants were transferred to clean tubes and ethanol fractions were evaporated via a speedvac vacuum concentrator (Thermo Savant SPD 2010) over 3 h at ramp level 5, without heat. The dried extracts were reconstituted in 250 µL of mobile phase solution, consisting of formic acid at 0.1% *v*/*v* in methanol and water (85:15), for the purposes of HPLC analysis. After homogenization, 200 µL of the extract was transferred to an HPLC vial with a 300 µL adaptor.

### 4.8. HPLC Detection Method for Cannabinoid Precursors

Olivetolic acid (CAS 491-72-5) and olivetol (CAS 500-66-3) standards were purchased from Santa Cruz Biotechnologies (Dallas, TX, USA). Extracts from *P. tricornutum* transconjugant strains and the wild type were then analyzed via HPLC (Agilent 1260 Infinity II) with a DAD detector. The column used was a ZORBAX Eclipse XDB-C18, 80Å, 4.6 × 250 mm, and 5 µm (Agilent; PN: 990967-902, Missisauga, ON, Canada). The sampler harvested and injected 10 µL of the sample into the system; the analysis conditions were the following: the temperature applied to the column was 30 °C, while the isocratic mobile phase comprised methanol at 85% and formic acid at 0.1%, mixed with 15% pure water at a flow rate of 0.4 mL per minute. The total run time of the analysis was 70 min per sample. The analyzed wavelengths were chosen based on the maximal peak of each standard (220, 230, and 280 nm).

### 4.9. Mass Spectrum Validation

All the LC-MS analyses in this study were performed at the Centre National en Électrochimie et en Technologies Environnementales, Inc. (CNETE, Shawinigan, QC, Canada). To validate the presence of the compounds of interest, the samples were analyzed under the same LC conditions but used the MS detector, achieving this by diluting each sample 10 times in methanol at LC-MS grade containing 0.1% formic acid and injecting the samples into a Thermo Scientific UPLC Dionex Ultimate 3000 MS LTQ XL UPLC-MS system equipped with a ZORBAX Eclipse XDB-C18 at 80 Å, 4.6 × 250 mm, and 5 µm (Agilent; PN.: 990967-902). The MS detector was set to analyze five scan events during the same period of analysis, taking one SIM event for each compound and one for large-scale TIC analysis (the event parameters are detailed in Appendix A, Figure A5). The analysis conditions used were the following: the temperature applied to the column was 30 °C and the isocratic mobile phase used comprised methanol 85% and formic acid 0.1%, mixed with 15% pure water at a flow rate of 0.4 mL per minute. The total run time of an analysis was 70 min per sample. The sample injection volume was 2.5 µL. The events observed at 220 nm were analyzed for 5 main events via the MS spectra corresponding to each of the main cannabinoids, while fragmentation was compared to the corresponding standard and theoretical fragmentation using the software program Mass Frontier 7.0.

### 4.10. Precursor Supplementation Assay

*P. tricornutum’*s empty vector (p*Pt*GE30) was grown for a full cycle from an established liquid culture until the OD_680_ was greater than 2 a.u. Afterward, six new subcultures were made with a starting inoculum of 0.1 OD_680_ in L1 media and then supplemented with 50 mg/L of zeocin for 9 days. The final volume of all subcultures was around 360 mL. Then, this volume was distributed as nine 30 mL cultures and placed into 50 mL Falcon tubes. The first set of triplicates was supplemented with ethanol as a negative control, the second set was supplemented with 1 mM of geranyl diphosphate ammonium salt (Sigma, CAS no 763-10-0) and 0.45 mM of olivetolic acid dissolved in L1 media, and the third set was supplemented with 0.27 mM of CBGA diluted in ethanol. Along with the p*Pt*GE30 subcultures, this assay was performed on at least two other recombinant strains containing at least one of the *Pt*OA cassettes. The Falcon tubes were returned to normal growth conditions for 16 h; the next day, each culture was centrifuged to separate the supernatant, then the pellet was washed and the washing liquid was kept for analysis. The metabolites were extracted from the cell pellets, as described in Section 4.7. The supernatant, washes, and cell extracts were analyzed via HPLC, and the signals of interest were confirmed later via MS analysis.

## 5. Conclusions

In conclusion, the bioengineering of *P. tricornutum* with cannabis genes represents a significant advancement in CB production technology. This innovative approach not only provides a sustainable alternative to traditional cannabis cultivation but also highlights the potential of marine microorganisms as versatile bioreactors for valuable natural products. As synthetic biology and metabolic engineering continue to evolve, the intersection of marine biology and cannabis biosynthesis could shape the future of CB production.

## Figures and Tables

**Figure 1 ijms-24-16624-f001:**
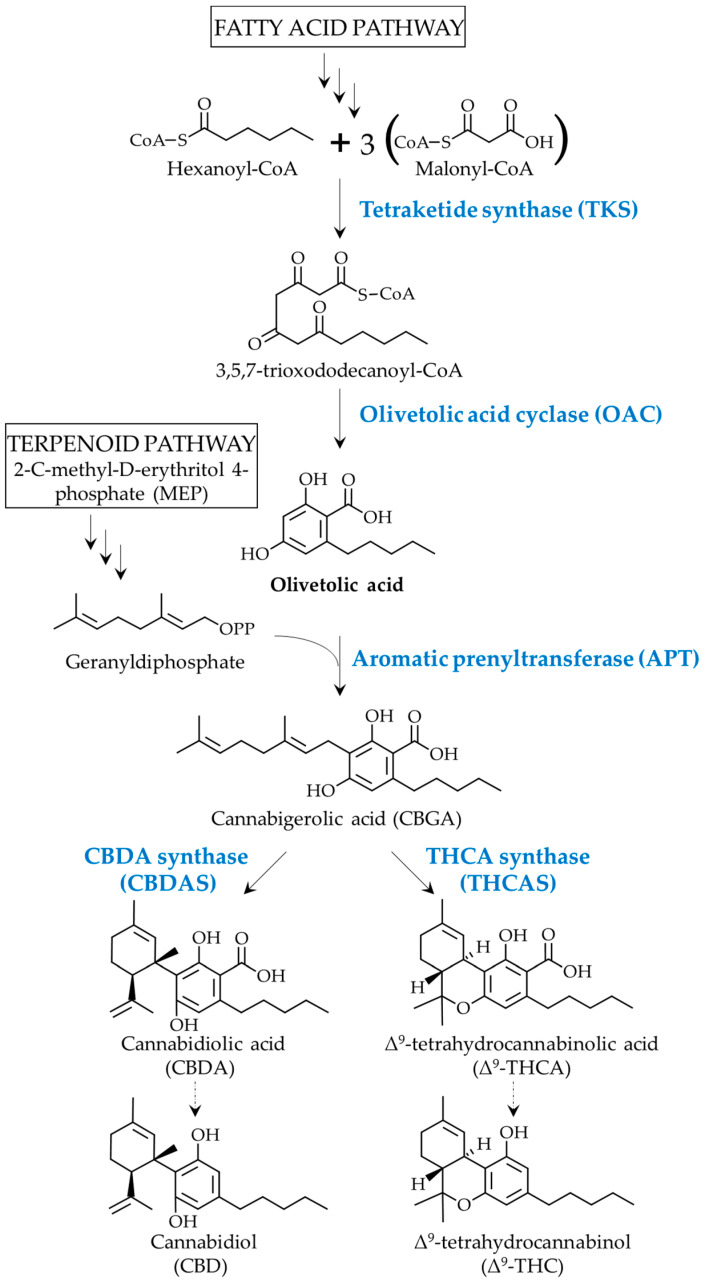
The proposed cannabinoid biosynthetic pathway, leading to the two major phytocannabinoids, THC and CBD. Enzymes are shown in blue. Bolded arrows represent an enzymatic reaction, whereas the dotted arrows represent non-enzymatic (heat or light) decarboxylation.

**Figure 2 ijms-24-16624-f002:**
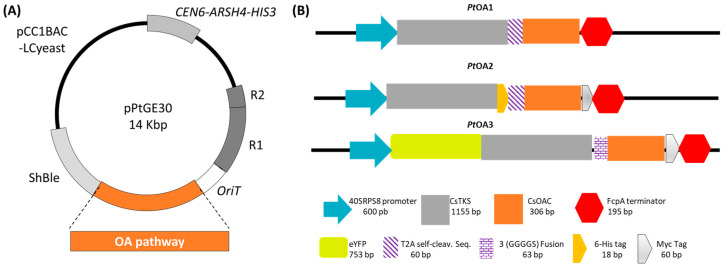
Design of the expression vectors *Pt*OA1, *Pt*OA2, and *Pt*OA3. (**A**) Schematic representation of the p*Pt*GE30 expression vector. (**B**) Description of the recombinant cassettes used to express *C. sativa* polyketide synthase (*Cs*TKS) and olivetolic acid cyclase (*Cs*OAC) under *P. tricornutum*’s constitutive promoter 40SRPS8, the fucoxanthin–chlorophyll binding protein A (FcpA) terminator, with or without tags (His = histidine, eYFP = enhanced yellow fluorescent protein, Seq = sequence, T2A = the *Thosea asigna* virus 2A cleavable sequence), and reporter genes of different sizes (base pair = bp).

**Figure 3 ijms-24-16624-f003:**
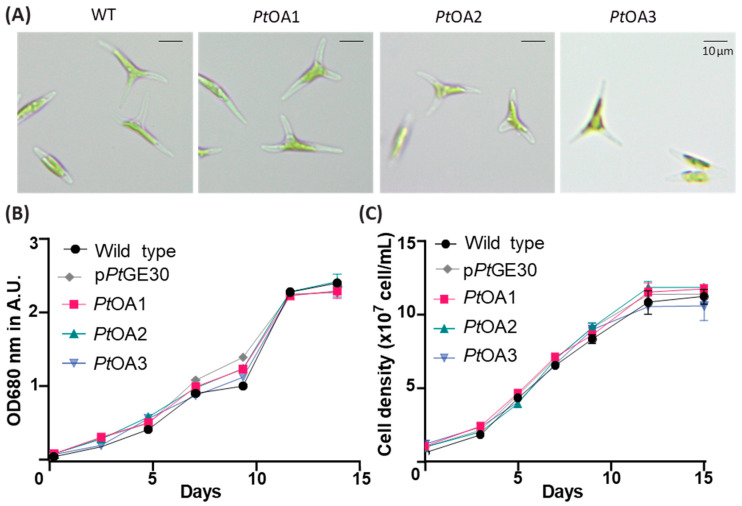
Cell shapes and growth curves of the transformed clones in comparison with wild-type *P. tricornutum*. (**A**) Optical microscope images of wild-type (WT) *P. tricornutum* and the transformed cells *Pt*OA1, *Pt*OA2, and *Pt*OA3. (**B**,**C**) *P. tricornutum* WT, transformed with pPtGE30, *Pt*OA1, *Pt*OA2, and *Pt*OA3 cell culture growth curves, followed by absorbance at 680 nm (**B**) or via cell count (**C**) during the 15 days of the cycle. The plotted values represent the means of three replicates and error bars represent the standard deviation.

**Figure 4 ijms-24-16624-f004:**
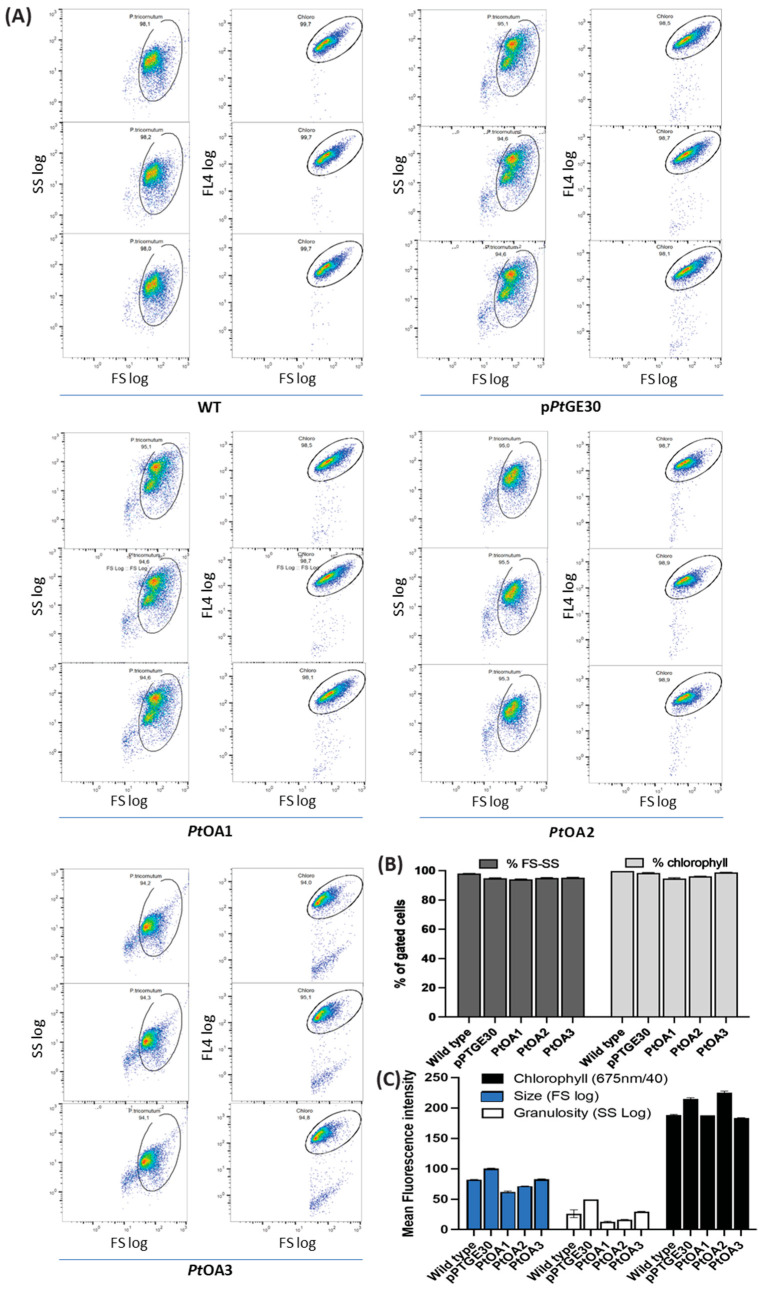
Flow cytometry observation for cell size and chlorophyll auto-fluorescence among *P. tricornutum* transconjugants and wild-type cultures. (**A**) Scatter plots for each strain and (**B**) a graphical representation of chlorophyll and size percentages among gated cells of wild-type *P. tricornutum* (WT), and *P. tricornutum*, harboring the empty vector p*Pt*GE30, *Pt*OA1, *Pt*OA2, and *Pt*OA3. (**C**) Histogram representing the mean chlorophyll fluorescence intensity, mean granulosity, and mean size scatter.

**Figure 5 ijms-24-16624-f005:**
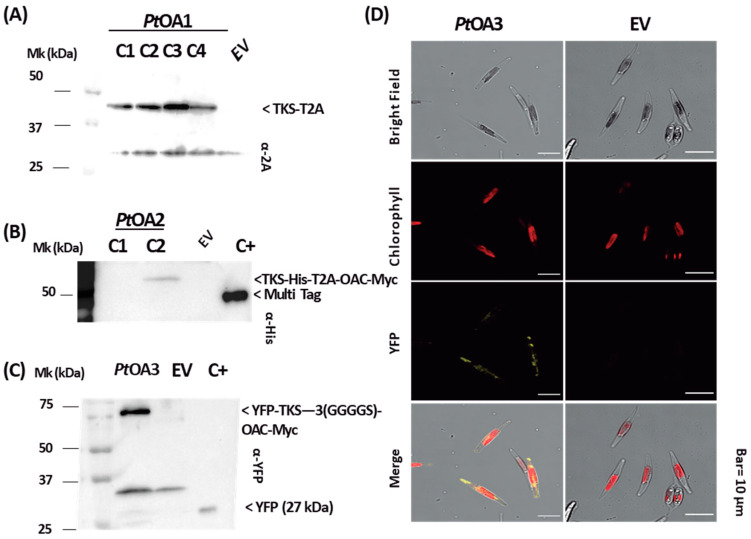
Expression and localization of heterologous proteins in transconjugant *P. tricornutum* strains. (**A**) Whole-cell extraction of four *Pt*OA1 (TKS-T2A-OAC) strains 1–4; TKS protein detection (45 kDa) was performed using anti-T2A antibodies. (**B**) Whole-cell extraction of two *Pt*OA2 (TKS-His-T2A-OAC-cMyc) clones 1 and 2; TKS:T2A:OAC protein detection (68 kDa) was performed using anti-His antibodies. (**C**) Whole-cell extraction of *Pt*OA3 (YFP-TKS-3(GGGGS)-OAC-cMyc) strains; YFP:TKS:OAC protein detection (85 kDa) was performed using anti-YFP antibodies. Mk: protein ladder; whole-cell extracts of the pPtGE30 culture were used as negative control (empty vector, EV); C+: Multi Tag was used as a positive control. All Western blots were obtained from 12% SDS-PAGE gel. For each construct, four clones were characterized and the ones showing protein production are presented in this figure. (**D**) Confocal microscopic images of *Pt*OA3 and pPtGE30 cells in a bright field, with autofluorescence of the chloroplast, YFP:TKS:OAC fluorescence, and the merging of three fields are shown.

**Figure 6 ijms-24-16624-f006:**
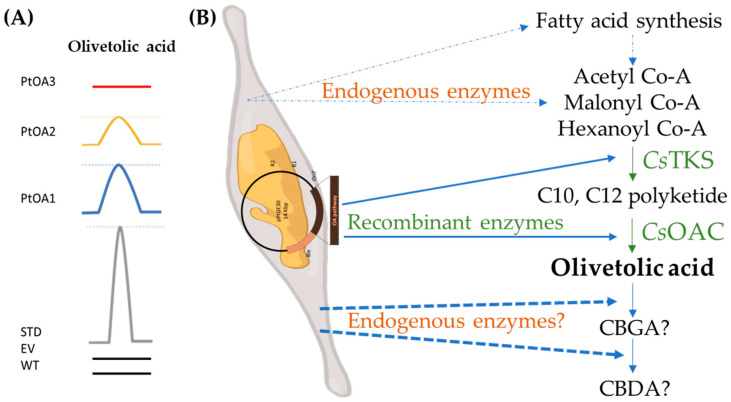
Olivetolic acid production, obtained via HPLC-UV analysis at 280 nm. (**A**) Curves represent the peak at the OA retention time of chromatograms, obtained following the injection of 10 µL of each extracted sample. At the bottom, a concentration of 10 μg/mL of commercial standards (STD) is shown: metabolite extraction from the *P. tricornutum* wild type (WT); *P. tricornutum,* transformed with an empty vector (EV = p*Pt*GE30), *Pt*OA1, *Pt*OA2, and *Pt*OA3. The peaks represent olivetolic acid. (**B**) Schematic representation of the bioengineered *P. tricornutum* for the production of cannabinoid precursors.

**Table 1 ijms-24-16624-t001:** Results of a supplementation assay of *P*. *tricornutum* with olivetolic acid (OA), geranyl diphosphate (GPP), and cannabigerolic acid (CBGA). The results show the mean detection levels that were found in three different strains of *P. tricornutum* transconjugants harboring p*Pt*GE30.

Supplemented mM			Detected			
OA	GPP	CBGA	HPLC	MS (*m*/*z*) Confirmation	HPLC	MS *(m*/*z*) Confirmation
0.45	1	0	CBGA	yes	CBDA, CBNA	No
0	0	0.27	CBDA	yes	THCA, CBNA	No

## Data Availability

Data is contained within the article.

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
