# Peer review of "Bioengineering of the Marine Diatom Phaeodactylum tricornutum with Cannabis Genes Enables the Production of the Cannabinoid Precursor, Olivetolic Acid"

_ijms, 2023, doi:10.3390/ijms242316624_

Round 1
Reviewer 1 Report
Comments and Suggestions for Authors
In the manuscript entitled “Bioengineering of the marine diatom Phaeodactylum tricornutum with cannabis genes enables the production of cannabinoid precursor, olivetolic acid”, authors could successfully express tetraketide synthase (TKS) and olivetolic acid cyclase (OAC) genes in the diatom P. tricornutum. Different techniques such as molecular, microscopic, proteomic, and biochemical approaches confirmed the production of TKS and OAC in the diatom. The topic of the manuscript is novel and original and it is designed very well with a good English style. I believe this manuscript is worth for publication and it could be attractive to many researchers. However, I have just one concern listed below;
1-Why did you use his-tag while you did not purify the recombinant protein? How did you interpret that the his-tag could decrease the T2a peptide functionality?
Comments on the Quality of English Language
English quality is good enough.
Author Response
RESPONSE: We wish to thank the reviewer for the comments to improve our manuscript which are truly appreciated.
Regarding the His-Tag, it was used to allow detection of the protein as specific antibodies (tailored to OAC or TKS) are extremely expensive. The use of His-tag allows us to follow the protein detection by western blot. Since the enzyme already been characterized, their purification was not done.
For the T2A peptide functionality, possible self-cleaving function decrease, it is only a hypothesis based on literature cases such as the ones referenced in the text showing that T2A efficiency is different in case the overall sequence of the transgene and the side sequences form a cis-acting element, or if the ribosome does not recognize the detaching site because of a structure that does not allow it. In both cases, we cannot be certain except if we repeat the experiment with different tagging sequences and evaluate the cleavage efficiency. Although an extremely interesting point, it was not the aim of our study.
(Chng, J.; Wang, T.; Nian, R.; Lau, A.; Hoi, K.M.; Ho, S.C.; Gagnon, P.; Bi, X.; Yang, Y. Cleavage Efficient 2A Peptides for High Level Monoclonal Antibody Expression in CHO Cells. MAbs 2015, 7, 403–412, doi:10.1080/19420862.2015.1008351. and Liu, Z.; Chen, O.; Wall, J.B.J.; Zheng, M.; Zhou, Y.; Wang, L.; Ruth Vaseghi, H.; Qian, L.; Liu, J. Systematic Comparison of 2A Peptides for Cloning Multi-Genes in a Polycistronic Vector. Sci Rep 2017, 7, 2193, doi:10.1038/s41598-017-02460-2.)

Reviewer 2 Report
Comments and Suggestions for Authors
The authors show the successful bioengineering of P. tricornutum with cannabis genes enabling the production cannabinoids using the natural metabolism of the algae for the synthesis of the necessary substrates for the cannabinoids syntheiszing enzymes.
The manuscript is well done and need just few adjustments: check the english grammar and style; check the reference style (species name in italics); ameliorate western blot pictures; describe all the abbbrevions in the pictures, e.g. EV, C+, etc.
Comments on the Quality of English Languagecheck the english all along the manuscript.
Author Response
RESPONSE: We are grateful and wish to thank the reviewer for the comments and suggestions to improve our manuscript which are truly appreciated.
1) We revised the whole manuscript and corrected english grammar and changed phrases to improve the style. Also, species names were revised and the ones that were not in italic in the reference list are now corrected.
2) EV, +C and other details are described fully in the figure title now. Thanks for pointing that out.
3) Regarding the western blot pictures, we modified the figure to increase the resolution and cropped the membrane to show specifically the bands of interest. The full uncropped images are in the supporting data Figure A2.

Reviewer 3 Report
Comments and Suggestions for Authors
Thank you for all the authors for the research showed in the manuscript. Despite the interest of the topic and the potential of P. tricornutum as a bioengineering tool, I wish to point some aspects to check before submit the final version of your manuscript.
-In figure number 1, please check that the size of the bonds are all the same, I am not sure of that.
- In 526 line, says that samples were resuspended in 60:40 water:acetonitrile and formic acid as mobile phase for HPLC whilst in 536 line says that the metabolites are separated using 85:15 methanol:water as mobile phase. This change of solvent mixture should be clarified.
-Figure A5 of supplemented material. Could you provide a better resolution images? They are pixel when I tried to zoom in.
Author Response
RESPONSE: We thank the reviewer for these pertinent comments, it will definitely help us improve the article. Modifications were made to the text and figure as suggested.
- All the bonds from Figure 1 are the same size except for one bond on the 3,5,7-triocodocecanoyl-CoA structure, where we elongated one bond to display the structure in a similar manner than the olivetolic acid. If we did not elongate that bond, the bottom vertical =O would show as a closed ring.
- We apologize for the confusion and we corrected the HPLC mobile phase composition in the revised manuscript. Thank you for pointing it out.
- Figure A5 was modified for a higher resolution.

Reviewer 4 Report
Comments and Suggestions for Authors In this manuscript, the authors present the successful integration of cannabis genesin Phaeodactylum tricornutum for the production of cannabinoid precursor.
They succeeded in obtaining P. tricornutum transconjugants that successfully produce
cannabinoid precursor. Moreover, the introduction of the studied genes led to the
synthesis of novel molecules with potential pharmaceutical applications.
The work of these authors is very interesting and exciting and opens many new areas.
The paper is beautifully written and contains a detailed and clear description of the
methods and results. I recommend publishing this work in the IJMS.
Author Response
RESPONSE: We would like to thank the reviewer for time and energy spent on reading this article, we are thrilled to read this positive review and to learn that you find this work interesting and exciting.
